EMBO
Molecular Medicine

# A novel fragile X syndrome mutation reveals a conserved role for the carboxy-terminus in FMRP localization and function

Zeynep Okray[1,2,3], Celine EF de Esch[4], Hilde Van Esch[2], Koen Devriendt[2], Annelies Claeys[1,2], Jiekun Yan[1,2], Jelle Verbeeck[1,2], Guy Froyen[1,2], Rob Willemsen[4], Femke MS de Vrij[5] & Bassem A Hassan[1,2,3,*]

## Abstract

Loss of function of the *FMR1* gene leads to fragile X syndrome (FXS), the most common form of intellectual disability. The loss of *FMR1* function is usually caused by epigenetic silencing of the *FMR1* promoter leading to expansion and subsequent methylation of a CGG repeat in the 5′ untranslated region. Very few coding sequence variations have been experimentally characterized and shown to be causal to the disease. Here, we describe a novel *FMR1* mutation and reveal an unexpected nuclear export function for the C-terminus of FMRP. We screened a cohort of patients with typical FXS symptoms who tested negative for CGG repeat expansion in the *FMR1* locus. In one patient, we identified a guanine insertion in *FMR1* exon 15. This mutation alters the open reading frame creating a short novel C-terminal sequence, followed by a stop codon. We find that this novel peptide encodes a functional nuclear localization signal (NLS) targeting the patient FMRP to the nucleolus in human cells. We also reveal an evolutionarily conserved nuclear export function associated with the endogenous C-terminus of FMRP. *In vivo* analyses in *Drosophila* demonstrate that a patient-mimetic mutation alters the localization and function of Dfmrp in neurons, leading to neomorphic neuronal phenotypes.

**Keywords** axon guidance; *Drosophila*; fragile X syndrome; nuclear export; nucleolus

**Subject Categories** Neuroscience; Genetics, Gene Therapy & Genetic Disease

## Introduction

Fragile X syndrome (FXS [MIM 300624]) is a highly prevalent, inherited disorder in humans causing intellectual disability accompanied by a spectrum of behavioral and physical abnormalities (Penagarikano *et al*, 2007). FXS patients typically show developmental delay and display an IQ below 70 and may suffer from significant decline in short-term memory, executive function, visuo-spatial abilities, and linguistic processing (Crowe & Hay, 1990; Belser & Sudhalter, 2001; Fisch *et al*, 2002; Cornish *et al*, 2004; Loesch *et al*, 2004). In most cases, the cognitive defects are accompanied by autistic behavioral phenotypes, including hyper-reactivity, social anxiety, aggression, stereotypic movements, mood disturbance, and attention deficiency (Lachiewicz & Dawson, 1994; Lachiewicz *et al*, 1994). Sleep disorders and vulnerability to epileptic seizures have also been reported in association with FXS (Berry-Kravis, 2002; Kronk *et al*, 2010). In the diagnosis of FXS, clinicians routinely screen for large expansions of the polymorphic CGG repeat elements in the *FMR1* locus (Xq27.3 [MIM 309550]). In FXS, the naturally occurring CGG repeats are expanded to numbers above 200, which usually leads to hypermethylation of the repeat itself and the upstream *FMR1* promoter (Fu *et al*, 1991; Oberle *et al*, 1991; Pieretti *et al*, 1991; Verkerk *et al*, 1991). As a consequence, the *FMR1* is transcriptionally silenced, and no protein product (FMRP) is formed.

FMRP is an RNA-binding protein that regulates many aspects of RNA biology, including RNA transport, stability and, most importantly, mRNA translation (Bagni & Greenough, 2005; Bassell & Warren, 2008). FMRP is ubiquitously expressed and is particularly abundant in the brain, ovaries and testes (Devys *et al*, 1993; Bakker *et al*, 2000), and a large number of potential FMRP mRNA and noncoding RNA targets exist (Fernandez *et al*, 2013). Collective evidence from mouse and fruit fly models indicates that local translational dysregulation in the absence of FMRP can impair early neuronal development, circuit formation, neurotransmission, and synaptic plasticity (Bassell & Warren, 2008; Zhang *et al*, 2001; Reeve *et al*, 2005).

Expansion of promoter-proximal CGG repeats and the consequent epigenetic suppression of *FMR1* expression is the leading

1  VIB Center for the Biology of Disease, VIB, Leuven, Belgium
2  Center for Human Genetics, University of Leuven School of Medicine and University Hospitals Leuven, Leuven, Belgium
3  Program in Molecular and Developmental Genetics, Doctoral School of Biomedical Sciences, University of Leuven, Leuven, Belgium
4  Department of Clinical Genetics, Erasmus Medical Center, Rotterdam, The Netherlands
5  Department of Psychiatry, Erasmus Medical Center, Rotterdam, The Netherlands
   *Corresponding author. Tel: +32 16 330752; E-mail: bh@kuleuven.be

genetic mechanism underlying FMRP deficiency in FXS patients. Deletions within or across the *FMR1* locus can also lead to the loss of FMRP, as reported in several fragile X case studies in the literature (Gedeon *et al*, 1992; Wohrle *et al*, 1992; Tarleton *et al*, 1993; Lugenbeel *et al*, 1995; Coffee *et al*, 2008). Although these two genetic mechanisms account for the majority of FXS patients, they do not usually yield insight into the functional properties of the different FMRP domains in a clinically relevant context. Pathogenic *FMR1* sequence variants that affect FMRP expression, localization, and function may also result in FXS. However, only a few potentially pathogenic point mutations in *FMR1* have so far been described (De Boulle *et al*, 1993; Collins *et al*, 2010a,b; Gronskov *et al*, 2011), and there is a need for detailed genetic and functional analyses to fully characterize these intragenic mutations. One notable exception that has yielded important information on the RNA-binding activity of FMRP is the I304N point mutation, identified in the *FMR1* coding sequence of a FXS patient with severe symptoms (De Boulle *et al*, 1993). This particular *FMR1* allele has been investigated extensively in numerous studies, and the mutation was found to profoundly alter many aspects of FMRP function via its effect on one of the RNA-binding domains called the K homology (KH) domain (Siomi *et al*, 1994; Feng *et al*, 1997; Tamanini *et al*, 1999; Laggerbauer *et al*, 2001; Schrier *et al*, 2004).

Despite its seemingly low occurrence in the FXS patient population (Collins *et al*, 2010b), searching and screening for potentially pathogenic *FMR1* sequence variants is essential. A segment of the patient population could otherwise remain undiagnosed and therefore may not benefit from future therapies. Moreover, elucidating the functional consequences of these mutations will provide the opportunity to study FMRP domains by identifying critical residues and characterizing the function of different domains in a clinically relevant context, as demonstrated by the I304N point mutation. The fact that genetic screens in the *D. melanogaster dfmr1* identified loss-of-function mutations in conserved residues (Reeve *et al*, 2008) lends credence to this notion.

Here, we report a novel *FMR1* frameshift mutation found in a patient with FXS symptoms. This mutation alters the open reading frame, creating a short novel amino acid sequence in the C-terminus followed by a premature stop codon. Functional characterization of this patient *FMR1* allele reveals that the mutation targets the protein to the nucleolus in cultured human and mouse cells. Furthermore, we observe the nuclear retention of the patient *FMR1* protein only when the C-terminus is truncated, hinting at the presence of a novel nuclear export function in the C-terminus of FMRP. A genetically versatile model for *FMR1* loss of function has been established in the fruit fly *Drosophila melanogaster*. The fly has a single homologue of *FMR1*, and its loss of function causes many phenotypes similar to those associated with FXS patients (Zhang *et al*, 2001; Morales *et al*, 2002; Pan *et al*, 2004; Reeve *et al*, 2005; Bassell & Warren, 2008; McBride *et al*, 2012). Using this model for *in vivo* analyses, we find that the NLS signal identified in the patient FMRP can target fly *dfmr1* protein to the nucleus in fly neurons, also in a C-terminus dependent fashion. This suggests that the nuclear export function of the FMRP C-terminus is evolutionarily conserved. Interestingly, the change in Dfmrp localization alters its function in neurons and leads to neomorphic phenotypes *in vivo*. Taken together, our results provide evidence for changes in FMRP regulation and function brought on by this novel fragile X patient mutation.

# Results

## Novel frameshift mutation identified in FXS patient

We selected 16 individuals from our in-house cohort of male patients with intellectual disability, for sequencing analysis of the *FMR1*. The selection was based on the presence of typical neuro-developmental features of fragile X syndrome including moderate intellectual disability (IQ < 60), autistic and/or stereotypic behavior, and impaired social interaction, associated with at least one of the following physical features: elongated face, macroorchidism, and/or large ears. The typical CGG repeat expansion in the *FMR1* locus was absent in all 16 males.

Among screened individuals, we found seven with previously unreported variations in the *FMR1* (Supplementary Fig S1A). In 5 of these individuals, single base polymorphisms were identified in intronic regions. In the other 2, single mutations were present in coding exons. In one of these individuals, we recovered a silent base substitution in exon 15. In the other, we identified a guanine insertion in exon 15 [1457insG] (Supplementary Fig S1A and B). This G-insertion mutation alters the open reading frame to one that is not used by any of the alternative *FMR1* isoforms (Ensembl Genome Browser), and is predicted to create a novel peptide sequence followed by a premature stop codon, which results in the truncation of the C-terminus of FMRP (Fig 1A). This modification also leads to the disruption of the RGG box, one of the three RNA-binding domains characterized in FMRP (Darnell *et al*, 2001; Ramos *et al*, 2003; Bagni & Greenough, 2005; Blackwell *et al*, 2010).

The *FMR1* allele with the G-insertion mutation was identified in a male patient with moderate to severe intellectual disability, first seen at the age of 36 years in a residential care setting (Fig 1B). He was born to healthy unrelated parents—now deceased—and has one healthy brother. He went to a specialized school for disabled children, and IQ measured at adult age was 41. Clinical examination at age 36 showed a man with normal build and growth parameters (50[th] percentile). He made almost no eye contact and had macroorchidism with bilateral testicular volume of 30 ml. His behavior was rather calm, yet periodic aggressive outbursts were noted. Overall, he presented with a typical fragile X clinical phenotype, although the CGG repeat number was reported normal (41 repeats) in multiple testing rounds.

## *FMR1* mRNA and protein levels are decreased in patient-derived cells

To determine whether *FMR1* mRNA and protein levels were affected by the G-insertion mutation, we established an immortalized EBV lymphocyte line from a blood sample of the patient and measured *FMR1* mRNA and protein levels in these cells. We found both to be significantly decreased (by ~60 and > 90%, respectively) compared to the levels in control cells (Fig 1C). We determined that the reduction of the *FMR1* mRNA in patient cells is primarily due to nonsense-mediated decay (NMD)—a translation-coupled endogenous mechanism to degrade faulty mRNAs with premature stop codons—since *FMR1* mRNA levels in the patient cells were restored to control levels upon treatment with the translational inhibitor puromycin (Fig 1C). Western blot analysis indicated that FMRP

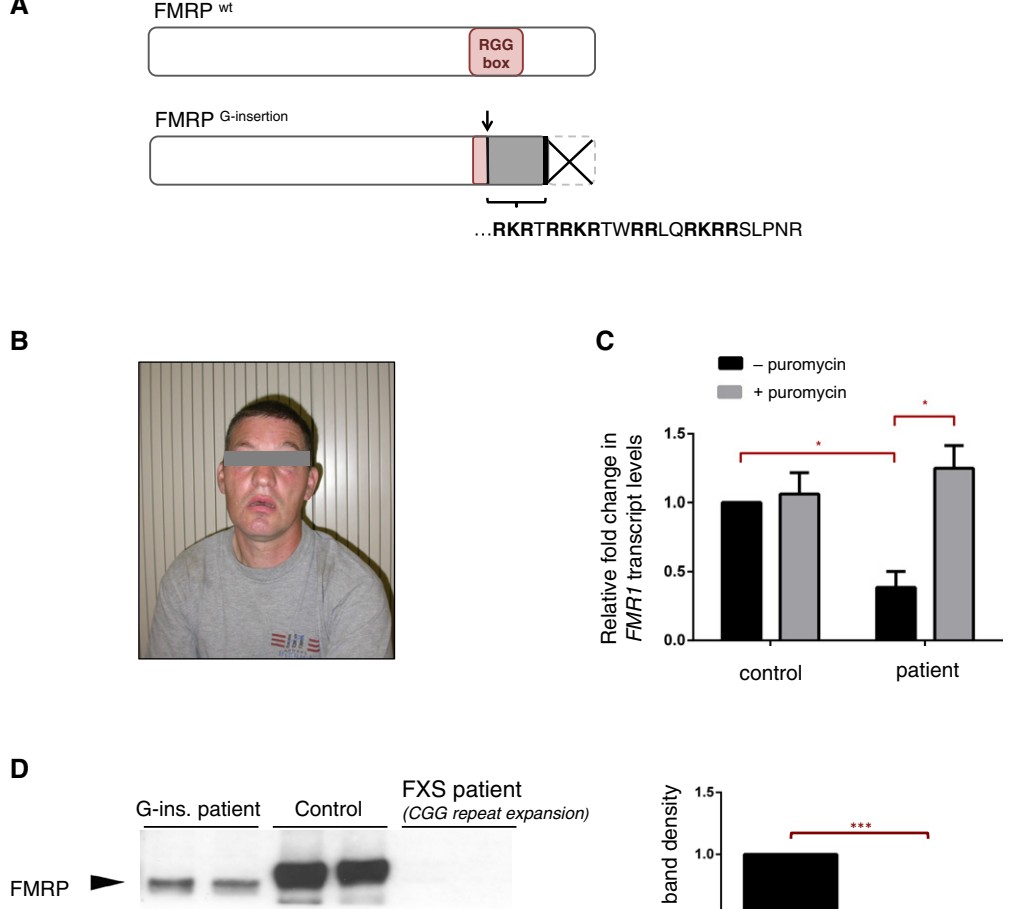

**Figure 1.  Novel G-insertion mutation identified in male with FXS symptoms.**

A   Predicted effects of the G-insertion mutation on FMRP are schematized. The insertion site corresponds to the beginning of the RNA-binding RGG Box, where it alters the open reading frame and thus disrupts the domain entirely. The frameshift creates a novel peptide sequence [RKRTRRKRTWRRLQRKRRSLPNR] followed by a premature stop codon, causing the truncation of the FMRP C-terminus.

B   Photograph of male patient with *FMR1^(G-ins.)* allele. Typical physical and behavioral features of FXS were noted in the patient, who shows moderate to severe intellectual disability.

C   *FMR1* mRNA levels were analyzed via RT–qPCR in EBV-transformed lymphocyte cells derived from the patient's blood samples. Patient cells showed a ~60% decrease in *FMR1* mRNA compared to control cells (*$P = 0.010$, $0.383 \pm 0.110$ SD, $n = 3$). Treatment with translational blocker puromycin restored *FMR1* mRNA levels in patient cells (*$P = 0.048$, $0.383 \pm 0.110$ SD versus $1.25 \pm 0.166$ SD, $n = 3$), suggesting that the reduction of *FMR1* mRNA in these cells is primarily due to nonsense-mediated decay.  RT–qPCR reactions were run in triplicate in three independent experiments. Fold changes in *FMR1* expression, normalized to *HRPT* expression, were calculated using the $\Delta\Delta C_T$ method, and analyzed statistically with a two-tailed *t*-test (GraphPad). Error bars represent mean values with SD.

D   Western blot analysis shows a significant decrease (> 90%) in FMRP protein levels in patient-derived cells (***$P = 0.008$, $0.13 \pm 0.031$ SD, $n = 2$), and the patient FMRP migrates slightly lower than the wild-type protein. Band intensities for the different cell lines were quantified, normalized for actin and analyzed statistically with a twot-tailed *t*-test (GraphPad). Error bars represent mean values with SD.

protein levels in patient-derived cells were significantly decreased compared to control cells from a healthy individual with no CGG repeat expansion in the *FMR1* locus (Fig 1D).

**Patient FMRP shows altered cellular localization**

To functionally characterize the patient FMRP protein, we began by expressing this novel allele in human cells. We transfected HEK293 cells with constructs expressing either GFP-tagged wild-type FMRP

(*GFP-FMR1^(WT)*) or GFP-tagged patient FMRP (*GFP-FMR1^(G-ins)*) under the control of a β-actin promoter. While wild-type FMRP was localized predominantly in the cytoplasm as expected, the patient FMRP showed a surprising localization pattern that was not observed with the wild-type protein (Fig 2A). Specifically, the patient FMRP forms bright nuclear inclusions with 100% penetrance. Given their size and appearance, we speculated that these inclusions might correspond to a nucleolar aggregation of FMRP. To test this idea, HEK293 cells expressing either *GFP-FMR1^(WT)* or

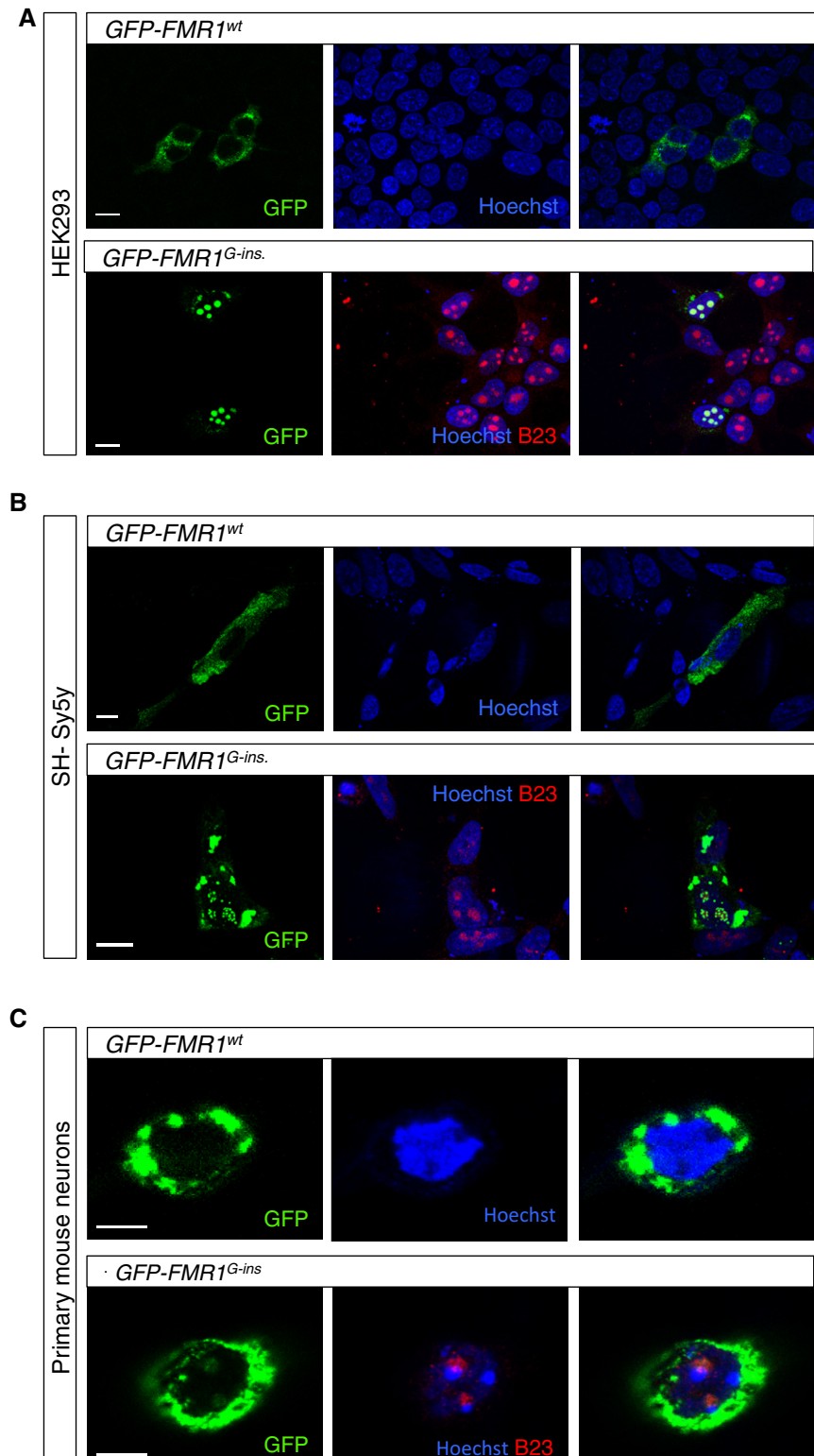

**Figure 2.  Patient FMRP aggregates in the nucleolus.**

A–C  HEK293 (A) and SH-Sy5y (B) cells, as well as primary neurons from *FMR1* KO mice (C), were transfected with vectors encoding the GFP-tagged wild-type FMRP (GFP-*FMR1*^WT) or the GFP-tagged patient FMRP (*GFP-FMR1*^G-insertion) under the control of a β-actin promoter. Wild-type FMRP localizes predominantly in the cytoplasm, whereas the patient FMRP forms nuclear and nucleolar aggregates that colocalize with nucleophosmin. Scale bars in (A, B) represent 10 μm. Scale bar in (C) represents 5 μm.

*GFP-FMR1^G-ins^* were stained for the detection of a nucleolar specific protein, nucleophosmin (NPM1). We found that the patient FMRP, unlike its wild-type counterpart, colocalizes with nucleophosmin, confirming the novel nucleolar localization of the patient FMRP. Finally, expression in SH-Sy5y human neuroblastoma cells (Fig 2B) as well as primary cultures of *FMR1*-KO mouse neurons resulted in a similar nuclear and nucleolar localization pattern for the patient FMRP, albeit at lower levels (Fig 2C).

### The frameshifted C-terminus contains a nuclear localization signal

We sought to explore the peptide changes in the patient FMRP that could target the protein to the nucleus. The G-insertion mutation and the accompanying frameshift lead to two significant modifications of the *FMR1* protein (Fig 1A): (i) the truncation of the C-terminus and (ii) the presence of a novel, 22 amino acid peptide sequence—both of which may contribute to the altered localization pattern. We

therefore dissected and characterized the contribution of each of the two components separately. We found that the C-terminus truncation (*GFP-FMR1^ΔCt^*) alone did not result in nuclear retention of FMRP (Fig 3A), suggesting that the novel amino acid sequence in the patient FMRP directs the change in localization observed. Interestingly, this novel peptide sequence is predicted to be a bipartite nuclear localization signal (NLS) when analyzed with the Scan-Prosite tool from Expasy (de Castro *et al*, 2006). This motif scanner identifies the NLS based on the adjacent stretches of Arg and Lys (Fig 1A). When we mutated adjacent Arg and Lys residues of the putative NLS to Ala (*GFP-FMR1^G-ins [NLS mutated]^*), the nuclear localization of the protein was no longer visible (Fig 3B). These results indicate that the novel peptide sequence in the patient FMRP is a functional nuclear localization signal. This was further confirmed by showing that the putative NLS could also target the cytoplasmic protein profilin to the nucleus (Supplementary Fig S2A). Importantly, the change in subcellular localization of the patient protein compared to the various controls does not correlate with levels of

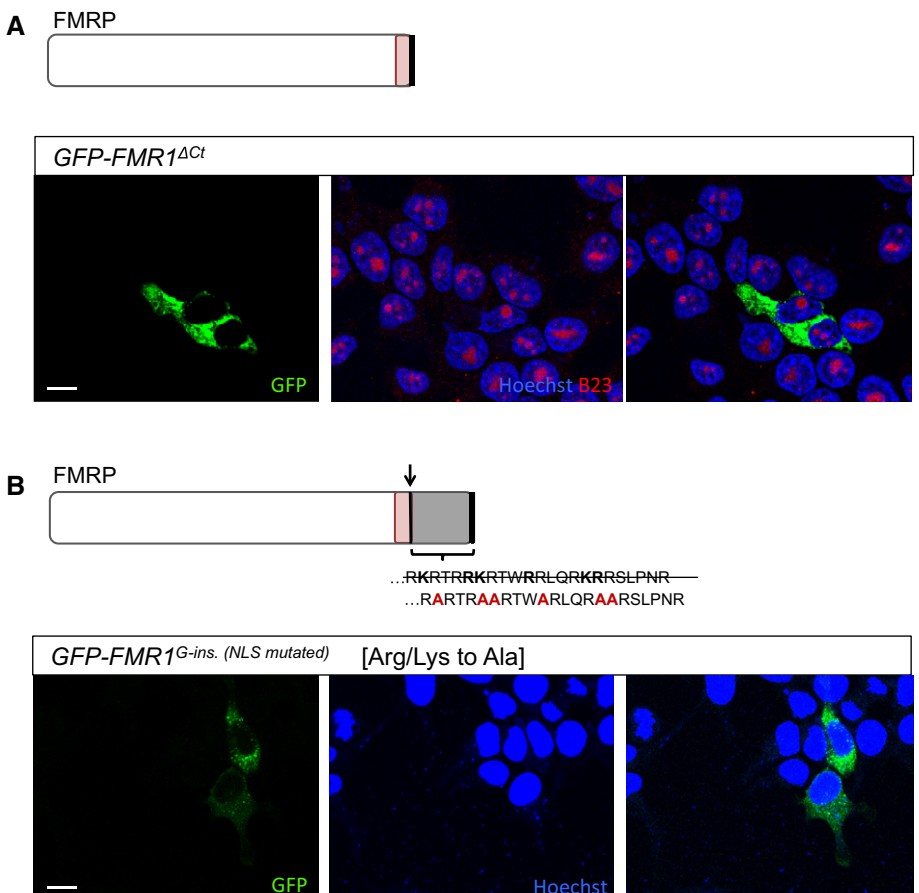

**Figure 3. Patient FMRP contains a novel nuclear localization signal in the C-terminus.**

A Transfected HEK293 cells expressing a GFP-tagged FMRP in which the C-terminus is truncated following the G-ins. mutation site (*GFP-FMR1^ΔCt^*). GFP-FMR1^ΔCt^ protein is exclusively cytoplasmic, suggesting that the truncation of the C-terminus alone is not sufficient to explain the localization change observed for the patient FMRP.

B Transfected HEK293 cells expressing a modified version of GFP-tagged patient FMRP (*GFP-FMR1^G-ins. [NLS mutated]^*). The novel amino acid sequence in the C-terminus of patient FMRP [RKRTRRKRTWRRLQRKRRSLPNR] contains stretches of arginines and lysines predicted to function as a nuclear localization signal (based on Expasy protein motif scanner, ScanProsite). Mutating adjacent R and K residues into alanines [RARTRAARTWARLQRAARSLPNR] abolishes the nucleolar localization of the patient FMRP, strongly suggesting that the novel amino acid sequence present in the patient FMRP C-terminus is a functional nuclear localization signal (NLS).

Data information: Scale bar represents 10 μm.

protein expression (Supplementary Fig S2B). Specifically, the patient mutation exhibits lower expression levels than the wild-type protein and the C-terminal truncation alone, but comparable levels to the controls where the NLS is mutated or the C-terminus is restored. Together, these data suggest that the nucleolar localization is due to the mutation and not to an increase in expression level.

**An intact C-terminus exports patient FMRP out of the nucleus**

Our results demonstrate that the frameshift caused by the G-insertion mutation creates a novel C-terminal peptide that targets the patient FMRP to the nucleus. However, it is not clear whether the truncation of the C-terminus contributes to the nuclear retention of the patient FMRP. To test this, we created an allele where the patient NLS motif is fused to the C-terminus of the full-length, wild-type $FMR1$ protein ($GFP\text{-}FMR1^{wt+NLS}$). Surprisingly, we did not observe any nuclear FMRP in cells expressing this specific $FMR1$ allele (Fig 4A). One possible explanation for this observation is that an intact C-terminus drives the efficient export of the patient FMRP from the nucleus. To test this, we studied the expression of $GFP\text{-}FMR1^{wt+NLS}$ in HEK293 cells treated with leptomycin B (LMB), a potent and specific inhibitor of nuclear export (Wolff et al, 1997). It has been shown that full-length wild-type FMRP does not show nuclear retention upon LMB treatment (Dury et al, 2013). In contrast, we find that nucleolar inclusions of the $GFP\text{-}FMR1^{wt+NLS}$ protein became visible upon treatment with LMB (Fig 4A), indicating that the presence of an intact C-terminus enables the nuclear export of FMRP bearing the patient NLS motif.

We also created another relevant variant of the patient $FMR1$ allele, where the frameshift caused by the G-insertion was restored—via the deletion of a single base pair—immediately before the premature stop codon ($GFP\text{-}FMR1^{G\text{-}ins[Ct\ revert]}$). At the protein level, the RGG box is almost entirely disrupted and replaced by the NLS sequence as in the case of patient FMRP, though the C-terminus truncation is avoided in this case. In line with our previous observations, we did not detect FMRP in the nucleus of cells expressing $GFP\text{-}FMR1^{G\text{-}ins[Ct\ revert]}$ (Fig 4B). However, treatment of these cells with LMB revealed nucleolar inclusions of FMRP (Fig 4B), once again demonstrating that the C-terminus truncation is critical for the nuclear retention of patient FMRP.

Taken together, our cell culture experiments demonstrate that the patient's G-insertion mutation leads to profound changes in the $FMR1$ protein product. The mutation leads to changes in subcellular localization mediated by the novel sequence and the truncation of the C-terminus. The data also suggest a potential nuclear export mechanism associated with the C-terminus of the FMRP protein in this context.

**Patient FMRP sequence changes cause novel neuronal phenotypes in vivo**

FMRP is a member of protein family known as the FXR protein family (Siomi et al, 1995; Zhang et al, 1995; Kirkpatrick et al, 2001). Although highly similar in their N-terminus, the C-terminus of the FXR protein family (Supplementary Fig S3) is not conserved at the sequence level (Kirkpatrick et al, 2001). We therefore wondered whether the effects of the mutation on the human protein are conserved in other FMRP homologs, or whether they are human specific. Therefore, we exploited the *Drosophila melanogaster* model, given its genetic tractability and success as a tool to study neurodevelopmental processes and related disorders (Okray & Hassan, 2013). The fruit fly has a single $FMR1$ homolog—$dfmr1$—that equally resembles human $FMR1$ and $FXR$ genes at the amino acid level (Wan et al, 2000; Morales et al, 2002). Various morphological, molecular, and behavioral phenotypes relevant to $dfmr1$ protein function have been described in *Drosophila* (Zhang et al, 2001; Morales et al, 2002; Pan et al, 2004; Reeve et al, 2005; Bassell & Warren, 2008; McBride et al, 2012). We selected a well-characterized neuronal population termed the Lateral Neurons ventral (LNv)—the fly circadian pacemaker neurons—whose connectivity phenotype is strongly affected by Dfmrp activity (Reeve et al, 2005). Specifically, the overexpression of $dfmr1$ in a wild-type background causes a consistent phenotype where the terminal axonal branches of sLNv neurons collapse (Reeve et al, 2005, 2008) (Fig 5A).

We took advantage of this robust assay and used the UAS-Gal4 binary expression system (Brand & Perrimon, 1993) to overexpress various $dfmr1$ alleles and examined changes in the $dfmr1$ gain-of-function phenotype in the LNv neurons. We created transgenic fly lines with different UAS-$dfmr1$ variants that dissect the different effects of the G-insertion mutation on the human FMRP protein (Fig 5B): wild-type $dfmr1$ ($dfmr1^{wt}$); C-terminus truncated $dfmr1$ ($dfmr1^{\Delta Ct}$); $dfmr1$ with the patient NLS only ($dfmr1^{wt+NLS}$); $dfmr1$ with both the patient NLS; and a truncated C-terminus mimicking the patient mutation ($dfmr1^{\Delta Ct+NLS}$). These variants were all inserted into the same genomic locus to minimize position effects on levels of transcription.

Overexpression of wild-type $dfmr1^{wt}$, $dfmr1^{\Delta Ct}$, and $dfmr1^{wt+NLS}$ leads to collapse of axonal branches in LNv neurons (Fig 6A and B). These findings suggest that there is no substantial loss in Dfmrp function associated with the truncation of the C-terminus alone, or with the presence of the NLS motif alone, despite significant differences in protein levels (Supplementary Fig S4). On the other hand, overexpression of $dfmr1^{\Delta Ct+NLS}$—where both modifications are present simultaneously—fails to induce the collapse of LNv axonal branches (Fig 6A and B). Instead, we detected novel axonal misguidance phenotypes with the overexpression of this allele, including aberrant bifurcations of axonal bundle termini, "tangles" of axons failing to extend medially, and misguided projections of single axons that appear to form loops (Fig 6C). Importantly, the *de novo* axonal phenotypes appear despite the fact that Dfmrp$^{\Delta Ct+NLS}$ is expressed at lower levels than the wild-type or Dfmrp$^{\Delta Ct}$ controls, and at comparable levels to the Dfmrp$^{wt+NLS}$ control (Supplementary Fig S4), consistent with the data for the human counterparts (Supplementary Fig S2B). Together, these data suggest that the defects caused by the Dfmrp$^{\Delta Ct+NLS}$ are an intrinsic property of the compound mutations, rather than expression levels of the mutant protein.

Finally, we investigated whether the neomorphic axonal phenotypes associated with $dfmr1^{\Delta Ct+NLS}$ correlate with a change in subcellular localization of the transgenic protein. We found that the Dfmrp which shows predominantly cytoplasmic localization in fly neurons (Wan et al, 2000), is targeted to and retained in the nucleus only when both the NLS motif is present and the C-terminus is truncated ($dfmr1^{\Delta Ct+NLS}$) (Fig 7A). Neither the truncation of the C-terminus ($dfmr1^{\Delta Ct}$) nor the presence of the NLS sequence alone ($dfmr1^{wt+NLS}$) is sufficient to retain Dfmrp in the nucleus (Fig 7B–D).

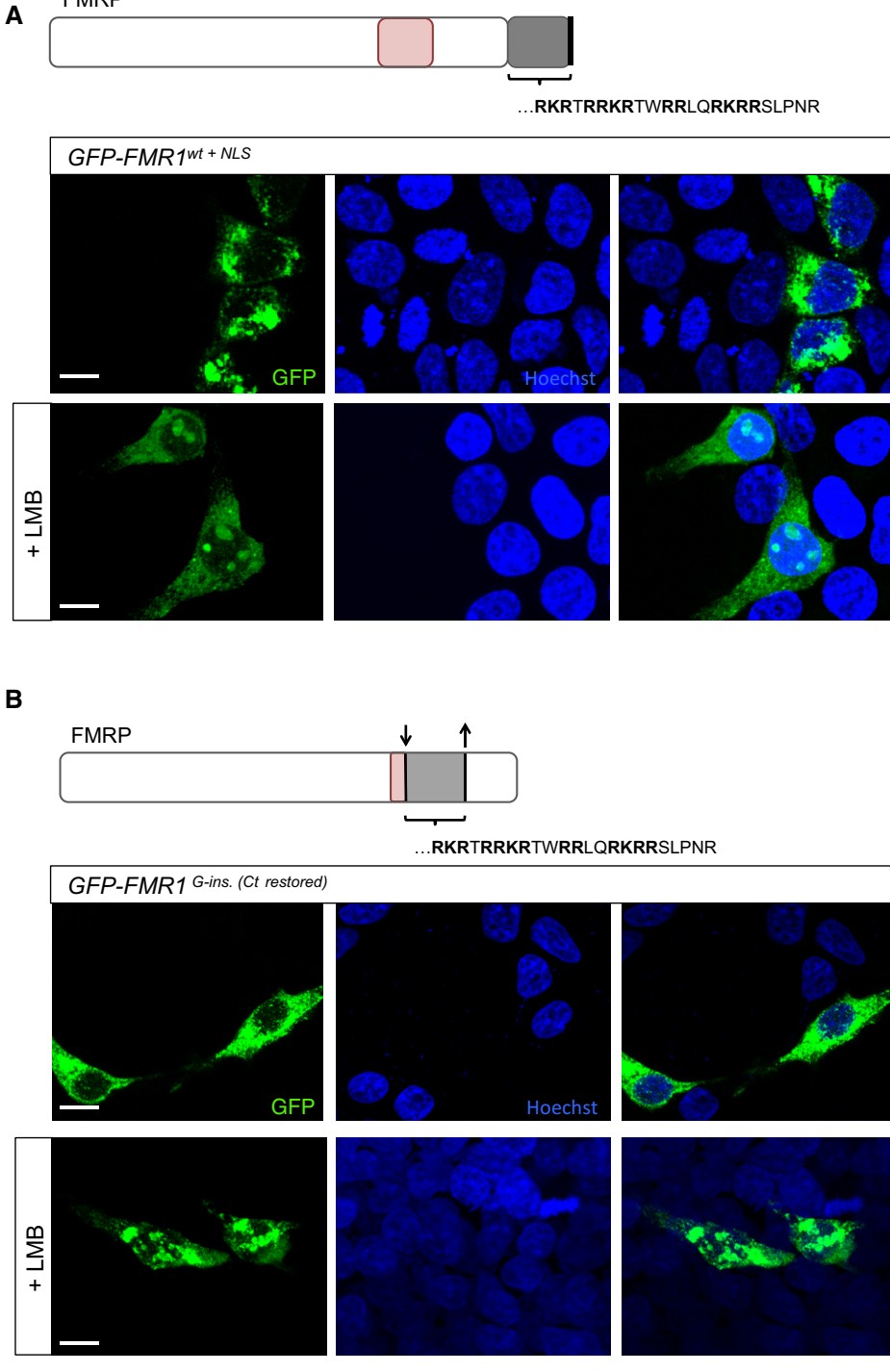

**Figure 4.  An intact C-terminus facilitates nuclear export of patient FMRP.**

A   Transfected HEK293 cells expressing GFP-tagged wild-type FMRP fused to the patient NLS motif (GFP-FMR1^wt+NLS). Unlike the patient FMRP, GFP-FMR1^wt+NLS protein is predominantly cytoplasmic and does not aggregate in the nucleus. However, treatment of HEK293 cells expressing GFP-FMR1^wt+NLS with leptomycin B—an inhibitor of nuclear export—resulted in the appearance of nucleolar inclusions. This suggests that the presence of a full-length C-terminus facilitates the nuclear export of FMRP in this context.

B   Transfected HEK293 cells expressing a modified version of GFP-tagged patient FMRP, in which the truncation of the C-terminus is reverted by restoring the open reading frame (*GFP-FMR1^G-ins.[Ct restored]*). The patient protein is not detected in the nucleus when the C-terminus is intact; however, nucleolar retention of the protein is observed upon leptomycin B treatment of the transfected cells. In line with results from (A), this suggests that an intact C-terminus enables nuclear export of the FMR1 protein.

Data information: Scale bar represents 10 μm.

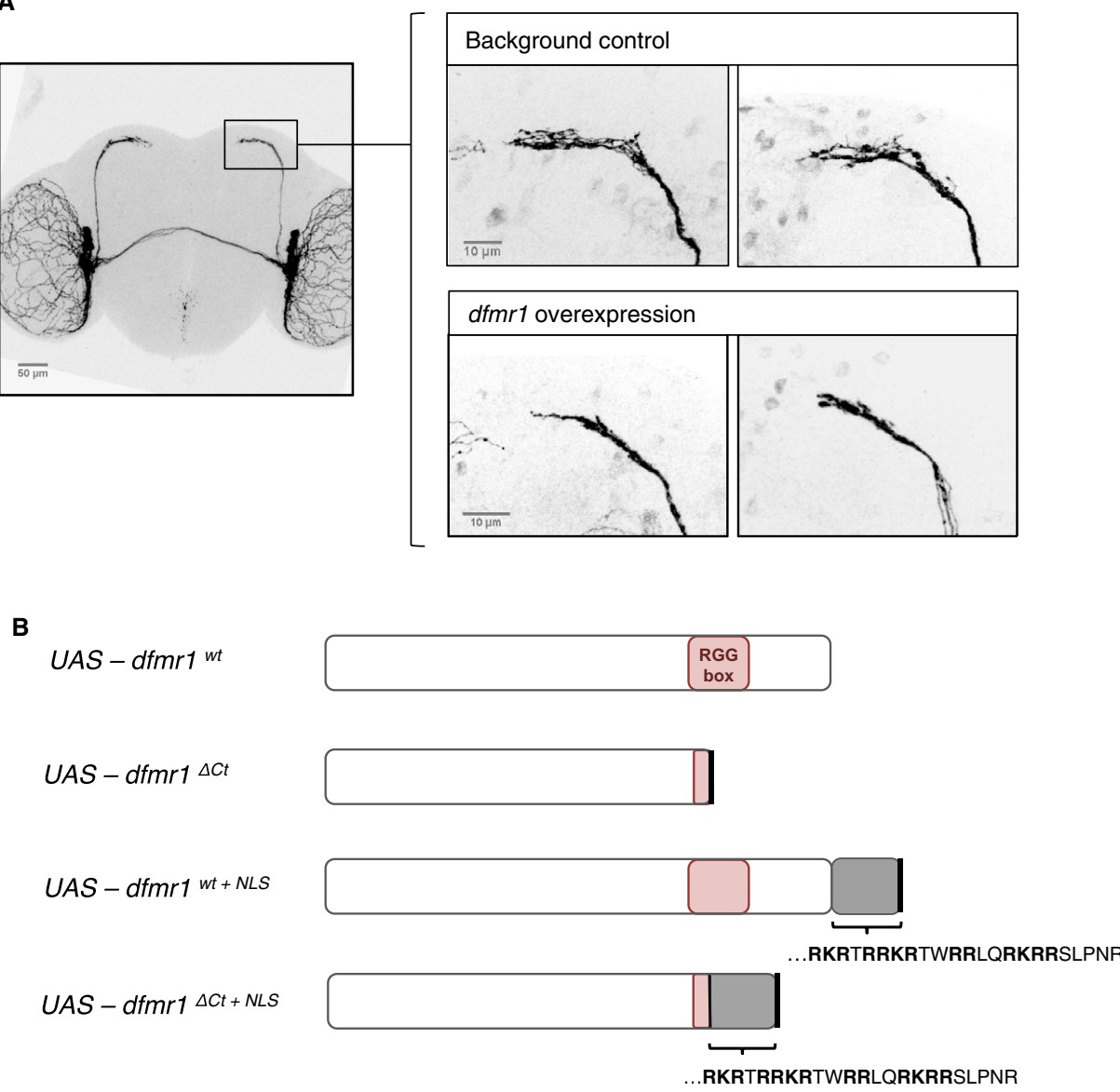

**Figure 5. sLNV neurons are sensitive to changes in Dfmrp function.**

A The morphology of small lateral ventral neurons (sLNvs) is sensitive to *dfmr1* activity. Overexpression of wild-type *dfmr1* in the sLNv neurons results in a consistent "axonal collapse" phenotype, where the branching of axonal termini of sLNvs is reduced. Scale bars represent 50 and 10 μm (magnified images).

B Transgenic fly lines were created bearing UAS-*dfmr1* variants, in order to assay *in vivo* functional changes associated with the effects of the patient mutation: the truncation of the C-terminus (*dfmr1*$^{\Delta Ct}$), presence of the patient NLS sequence (*dfmr1*$^{wt+NLS}$), or both (*dfmr1*$^{\Delta Ct+NLS}$).

Altogether, these data suggest that the gain of function observed for the patient-like form of Dfmrp is linked to the nuclear retention of the protein. Consistent with this idea, further increasing expression of Dfmrp using two copies of UAS-*dfmr1* does not cause nuclear localization nor axonal looping or bifurcation (Supplementary Fig S5).

## Discussion

Here, we have identified and characterized the effects of a novel, intragenic *FMR1* frameshift mutation discovered in a patient with typical FXS symptoms. To our knowledge, the G-insertion mutation reported here is the first naturally occurring, clinically relevant mutation to significantly alter the localization of FMRP.

The frameshift leads to profound changes in the peptide sequence: A premature stop codon results in the truncation of the C-terminus, abolishing the RNA-binding RGG Box, and creates a novel amino acid sequence encoding a nuclear localization signal (NLS). This NLS sequence can target the FMRP protein to the nucleolus. Interestingly, restoring the C-terminus enables efficient nuclear export of the protein in this context. Finally, using a *Drosophila* model, we show that the presence of the NLS sequence together with the truncation of the C-terminus alters FMRP function in neurons *in vivo*.

**A**    *Pdf-Gal4, UAS-CD8-GFP*  x  …

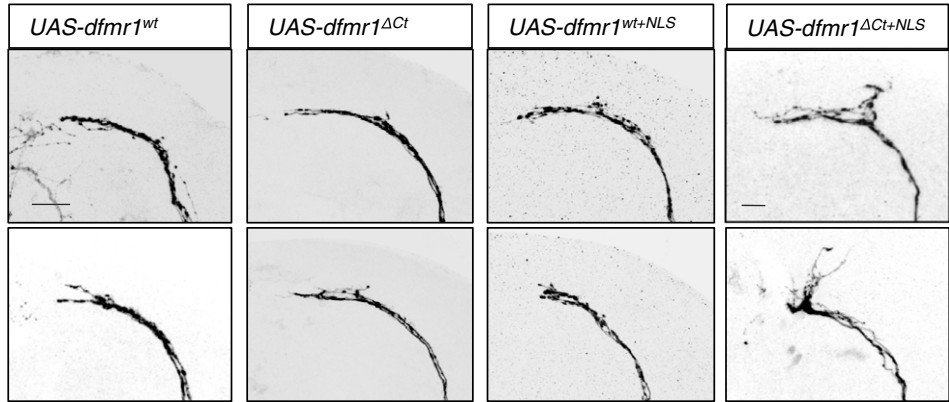

**B**

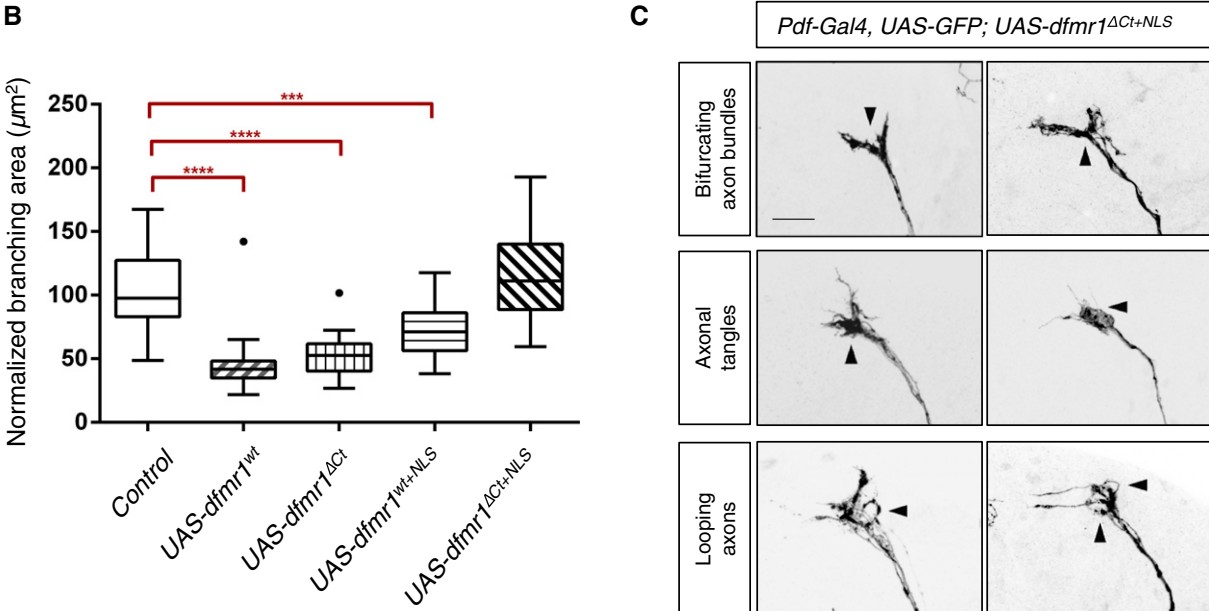

**C**    *Pdf-Gal4, UAS-GFP; UAS-dfmr1$^{\Delta Ct+NLS}$*

**Figure 6. Compound effects of the patient mutation confer a neomorphic function for Dfmrp in sLNV neurons.**

A   Overexpression of wild-type *dfmr1* (*UAS-dfmr1$^{wt}$*) using a LNv neuron-specific driver line (*Pdf-Gal4*) causes reduction of branching area—"axonal collapse"—in the sLNv axonal termini. *dfmr1$^{\Delta Ct}$* and *dfmr1$^{wt+NLS}$* overexpression phenocopies overexpression of *dfmr1$^{wt}$*, suggesting that the truncation of the C-terminus or the presence of the patient NLS alone does not impair or alter the functional efficacy of *dfmr1* in this context. However, overexpression of *dfmr1$^{\Delta Ct+NLS}$* fails to cause collapse of sLNv termini, indicating a significant loss or change of Dfmrp protein function. Axonal morphology is visualized by expressing CD8-GFP under the control of the same neuronal driver. Scale bar represents 10 μm.

B   Quantification of axonal termini branching in LNv neurons overexpressing various *dfmr1* alleles. Overexpression of *dfmr1$^{wt}$*, *dfmr1$^{\Delta Ct}$*, or *dfmr1$^{wt+NLS}$* significantly reduced axonal branching area [****$P < 0.0001$; control $101.3 \pm 29.76$ SD, $n = 30$; *UAS-dfmr$^{wt}$* $45.41 \pm 21.69$ SD, $n = 29$; *UAS-dfmr$^{\Delta Ct}$* $52.31 \pm 14.67$ SD, $n = 32$; *UAS-dfmr$^{wt+NLS}$* $72.45 \pm 21.83$ SD, $n = 29$; *UAS-dfmr$^{\Delta Ct+NLS}$* $117.8 \pm 36.96$ SD, $n = 28$]. *dfmr1$^{\Delta Ct+NLS}$* overexpression does not result in a significant change of axonal branching area compared to the background control [Pdf-Gal4, UAS-GFP] ($P = 0.672$). For quantification purposes, each branching area ($n$) was manually outlined, starting from the first point of defasculation of the axonal bundle. These measurements were normalized for variability in brain size across samples using LNv commissure length measurements for each brain. Two-tailed $t$-tests using Welch's correction were then used to compare controls with mutant phenotypes (GraphPad). Error bars represent mean values with SD. Dots visible for box plots of *UAS-dfmr$^{wt}$* and *UAS-dfmr$^{\Delta Ct}$* represent samples that outlier the whiskers.

C   Overexpression of *dfmr1$^{\Delta Ct+NLS}$* results in aberrant axonal termini morphology. Axonal guidance defects were frequently observed (arrowheads) and scored manually for bifurcations of the axon bundle (9/48), tangling of axons in the termini (13/48), and apparent looping of axons (7/48).

Our findings strongly support the notion that genetic mechanisms other than CGG repeat expansions and deletions in the *FMR1* locus can underlie fragile X syndrome. Although it is likely that the patient's symptoms arise from the decrease in FMRP levels, it cannot be ruled out that impaired and aberrant functions associated with the remaining mutant protein also contribute. It is worth noting that males with reduced FMRP levels yet normal range IQs have been described, indicating that reduction—as opposed to total

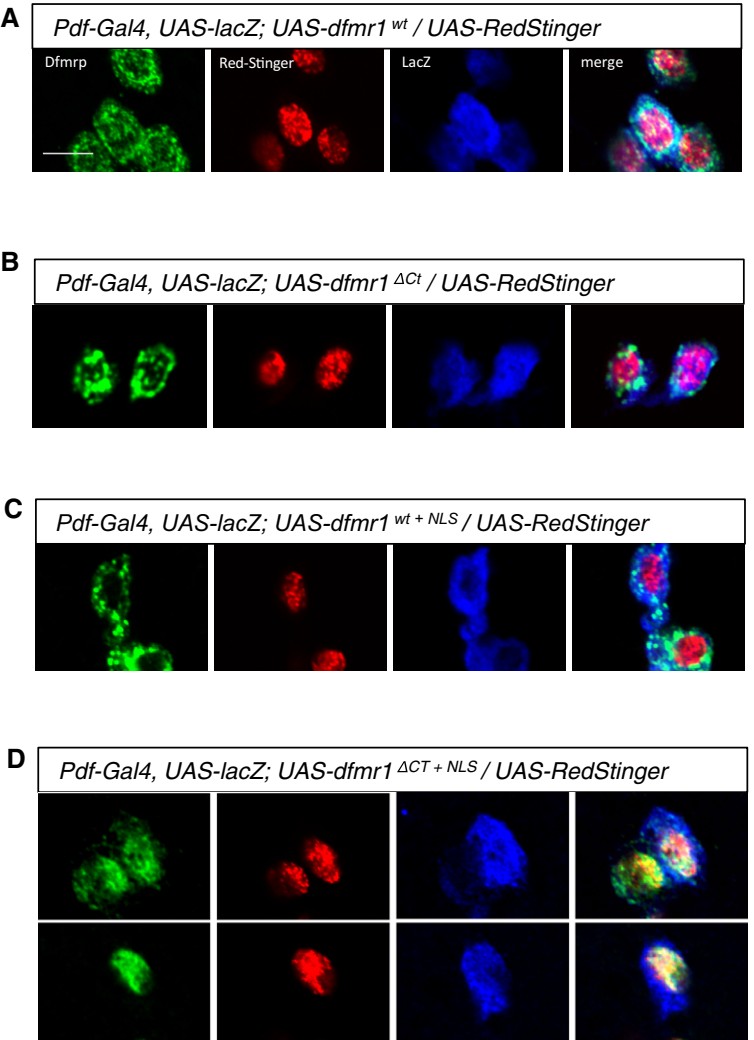

**Figure 7.  Subcellular localization of the Dfmrp variants in sLNv neurons.**

A–D  Subcellular localization of transgenically expressed *dfmr1*wt, *dfmr1*ΔCt, *dfmr1*wt+NLS, and *dfmr1*ΔCt+NLS proteins in LNv neurons. DfmrpΔCt+NLS shows nuclear localization (A), colocalizing with the genetically encoded nuclear marker RedStinger, while Dfmrpwt, DfmrpΔCt, and Dfmrpwt+NLS are predominantly cytoplasmic (B–D). These results suggest that an intact C-terminus can mediate the nuclear export of the Dfmrp protein. Scale bar represents 5 μm.

absence—of FMRP is not necessarily always causal to FXS (Hagerman *et al*, 1994).

The striking change in localization observed for the patient FMRP appears to confer a change in function for the protein. The nucleolar aggregation and retention of the mutant protein could lead to an exaggeration of a previously proposed (Willemsen *et al*, 1996) and recently confirmed (Taha *et al*, 2014) molecular function for FMRP: Specific endogenous isoforms have been detected in trace amounts in the nucleolus, where they biochemically interact with nucleolin, a multi-functional nucleolar protein required for rRNA transcription and several steps of ribosome biogenesis. Despite its subtlety, it is conceivable that localization of endogenous FMRP to the nucleolus has significant functional consequences. For example, it is proposed that the combined action of FMRP and nucleolin in this context can potentially impact ribosomal biology (Kim *et al*, 2009; Taha *et al*, 2014). The patient *FMR1* allele identified in this study may provide a unique opportunity to gain insight into this unexplored nucleolar function of FMRP. It is worth noting that in a sense the patient mutation mimics a reduction of the diversity of FMRP isoforms to a single, predominantly nucleolar form. Future experiments at endogenous expression levels *in vivo* will help ascertain whether these interpretations are true.

Interestingly, the mammalian paralogs of *FMR1*—the *FXR1* [MIM 600819] and *FXR2* [MIM 605339] protein products—have been shown to shuttle between the cytoplasm and nucleolus (Tamanini *et al*, 1999, 2000). Moreover, cells appear to regulate the nucleolar shuttling of the *FXR1* protein by generating multiple isoforms with different C-termini, where some isoforms of *FXR1* generate a C-terminal nucleolar localization signal, as a result of a frameshift induced by alternative splicing of the 3′ end of *FXR1* mRNA (Tamanini *et al*, 2000). The fact that the C-terminus is highly variable across *FMR/FXR* proteins might suggest that changes in the C-terminus underlie the functional diversification of the protein family.

Our findings also highlight the possibility that the FMRP C-terminus regulates nuclear/nucleolar shuttling of the endogenous protein. Nuclear shuttling of FMRP has been widely studied, although the exact mechanisms and protein motifs involved still remain somewhat unclear (Kim *et al*, 2009). Despite the fact that the C-terminus is highly divergent across *FMR1* homologs (Wan *et al*, 2000; Kirkpatrick *et al*, 2001), our data suggest that the nuclear export function mediated by this domain is evolutionarily conserved.

At any rate, the unique effects of the G-insertion mutation on the *FMR1* protein suggest that future molecular and functional analyses of the patient allele identified herein can yield crucial insight into FMRP function in a clinically relevant context.

# Materials and Methods

### Ethical considerations and patient consents

The clinical screening protocol was approved by the appropriate Institutional Review Board of the University Hospitals of Leuven (Belgium), which operates in agreement with the principles in the WMA Declaration of Helsinki. Informed consent was obtained from the parents/guardians of the affected patients, and permission to publish photos of the patient was granted. Mouse housing conditions and experiments were approved by the Dutch Ethical Committee (DEC) under Erasmus MC Permit Number EMC 140-09-06.

### Data deposition

The patient FMR1 variant has been submitted to the Leiden Open Variation Database (LOVD), with the accession ID #00025860.

### CGG repeat analysis and *FMR1* sequencing

Genomic DNA from patients was isolated from peripheral blood according to standard procedures and stored at 4°C. Molecular *FMR1* CGG repeat expansion analyses were performed with an in-house PCR method and with Southern blot, using probe StB12.3 after a combined EcoRI and EagI digestion, as described in Rousseau *et al* (1994). *FMR1* locus was amplified from genomic DNA for Sanger sequencing, performed by VIB Genetic Service Facility (University of Antwerp, Belgium).

### Culture and treatment of EBV-transformed lymphoblastoid cell lines

EBV-transformed lymphoblastoid cell lines were generated from patient peripheral blood using standard protocols. Cells were propagated as a suspension culture in DMEM/F12 medium containing 10% fetal bovine serum. For experiments involving puromycin, cells were treated with 200 μg/ml puromycin (Sigma-Aldrich) overnight (15 h). Cells were washed twice with PBS before RNA extraction.

### RNA extraction and RT–qPCR

Total RNA was extracted using TRIzol reagent (Life Technologies), following the manufacturer's protocol. The RNA extract was cleaned up using RNeasy kit (Qiagen). One μg of total RNA was used for cDNA synthesis (Quantitect Reverse Transcription kit, Qiagen), providing template for the qPCR. qPCR mixes were prepared using LightCycler 480 SYBR Green I Master kit (Roche Life Science), following the manufacturer's instructions. The qPCR was carried out using the Roche LightCycler 480 Real-Time PCR System, and software recommended standard 3-step cycles were used (95°C, 10 s; 60°C, 10 s, and 72°C, 10 s) for 40 cycles. Reactions were run in triplicate in three independent experiments, where each experiment was analyzed independently, with control (−puromycin) *FMR1* levels were set to 1. The mean of expression values of the housekeeping gene *HPRT* was used as an internal control to normalize for loading variability. Fold changes in *FMR1* expression were calculated using the $\Delta\Delta C_T$ method and analyzed statistically with a two-tailed *t*-test (GraphPad). Error bars represent mean values with SD.

### Western blotting

Protein from total cell lysates was resolved in NuPAGE 4–12% Bis–Tris polyacrylamide gels (Life technologies) under denaturing conditions and transferred to nitrocellulose membranes (Whatman, GE Healthcare Life Sciences). The blots were probed using anti-FMRP antibody (1C3 MAB2160 Millipore), diluted 1:500 in 3% bovine serum albumin, anti-GFP 1:750 (AB3080; Millipore), and anti-actin (JLA20 Hybridoma bank or MAB1501 Chemicon) diluted 1:500 or 1:10,000, respectively, in 5% milk. Anti-Dfmrp (5D7) antibody was specifically generated by EMBL Monoclonal Antibodies Core Facility and used in 1:500 dilution in 5% milk. ECL IgG horseradish peroxidase-linked antibodies (Amersham, GE Healthcare Life Sciences) or Li-Cor IRDye antibodies were used as secondary antibodies. Bands were visualized using the ECL Western Blotting Detection System (GE Healthcare Life Sciences) or the Odyssey system (Li-Cor). Band intensities for the different cell lines were quantified using ImageJ software, normalized for actin and analyzed statistically with a two-tailed *t*-test (GraphPad). Error bars represent mean values with SD.

### Cloning

We acquired a plasmid in which the *GFP-FMR1^wt* construct was cloned in a standard mammalian expression vector under a beta-actin promoter (Levenga *et al*, 2009). We used this plasmid as template for generating all *FMR1* variant alleles. We modified the *FMR1^wt* insert using site-directed mutagenesis (QuickChange II Site-directed Mutagenesis Kit, Stratagene) or (overlap extension) PCR (Phusion polymerase, NEB) with classical restriction enzyme cloning.

The fly overexpression constructs were created using the pUAST-attp-vector backbone. The *dfmr1* variants were cloned from template wt *dfmr1* cDNA. We modified the wild-type *dfmr1* insert by using (overlap extension) PCR (Phusion polymerase, NEB) with classical restriction enzyme cloning.

### Cell culture and transfection experiments

Human embryonic kidney (HEK) 293T cells and human neuroblastoma SH-SY5Y cells were cultured in Dulbecco's modified Eagle's medium (DMEM) (Lonza, Verviers, Belgium) supplemented with

10% fetal bovine serum, 1% penicillin, and 1% streptomycin at 37°C in a 5% $CO_2$ humidified incubator. The cell lines used were standard and tested regularly for mycoplasma contamination and never found positive. Primary hippocampal neurons of *Fmr1* KO(2) mice were prepared and cultured as described in de Vrij *et al* (2008). For each experiment, one pregnant *Fmr1* KO(2) female mouse in the C57Bl/6 background was sacrificed, when embryos were at day E17. Neurons were then pooled from all embryos per litter, with average litter size being 7. The mice were housed at the Erasmus MC animal facility (Rotterdam, the Netherlands), under standard housing and husbandry conditions, approved by the local animal welfare committee.

Cultured cells were transfected with the appropriate expression constructs using polyethylenimine (PEI) (Polysciences Inc., Warrington, PA, USA) for 293T cells and Lipofectamine 2000/Plus Reagent (Invitrogen) for SH-SY5Y cells and 21-day-old primary mouse neurons according to standard manufacturers protocols. One day after transfection, cells were fixed with 4% formaldehyde, washed in PBS, and then washed in a Hoechst solution (0.67 mg/ml) (Invitrogen) before mounting in Mowiol mounting solution (Mowiol 4–88) after a final PBS wash step. For visualization of nucleoli, cells were incubated overnight with primary nucleophosmin (NPM1) antibody (Santa Cruz SC-6013) 1:100 in staining buffer containing 0.05 M Tris, 0.9% NaCl, 0.25% gelatin, and 0.5% Triton X-100, pH 7.4 at 4°C, followed by standard incubation with secondary anti-goat Cy3 1:200 (Jackson Immunoresearch). Then, the cells were fixed, stained with Hoechst and mounted in Mowiol. Imaging was done using a Leica SP5 confocal microscope and LAS AF lite software (Leica Microsystems). For the leptomycin B (LMB) experiments, 293T cells were seeded in 12-well plates and transfected with the appropriate constructs using PEI as well. Two days after transfection, 50 ng/ml LMB (Sigma) was added to the cells for four hours. For cell culture experiments, wild-type and mutant FMRP constructs were transfected in parallel, in duplicate, in at least three independent experiments. Cells were transfected in random order. During imaging, the experimenter was blinded to the transfection conditions.

## Fly stocks and husbandry

Flies were raised at 25°C, in standard rearing conditions. Following fly strains were used for experiments:
yw; Pdf-Gal4, UAS-CD8-GFP; UAS-CD8-GFP (Ayaz *et al*, 2008)
w;; UAS-dfmr1$^{wt}$/TM6C
w;; UAS-dfmr1$^{\Delta Ct}$/TM6C
w;; UAS-dfmr1$^{wt+NLS}$/TM6C
w;; UAS-dfmr1$^{\Delta Ct+NLS}$/TM6C
Canton S 10 (*w*)
w;Pdf-Gal4 UAS-lacZ/Cyo;
UAS-RedStinger (Bloomington #8545)
Elav$^{C155}$-Gal4 (Bloomington #458).

The *UAS-dfmr1* strains were created using PhiC31 mediated transgenesis in the VK33 docking site (3L, 65B2) (Venken *et al*, 2006). Injection of the embryos was done in-house.

## Immunochemistry on fly tissue

Brains were dissected during morning hours (with genotypes in random order) from 0- to 7-day-old adult flies and stained with

### The paper explained

#### Problem
Fragile X syndrome (FXS) is the most common genetic cause of intellectual disability and includes features such as autistic-like behaviors, and distinctive craniofacial phenotypes. FXS is almost always caused by epigenetic silencing of the *FMR1* gene. While the FMRP protein is heavily studied, there is only a single confirmed disease-causing point mutation reported thus far, making the link between functional analysis of the FMRP domains and the disease difficult to analyze.

#### Results
We sequenced patients with FXS symptoms but no epigenetic silencing of *FMR1*. We found a point mutation in one patient, which causes a frameshift in the FMRP sequence and a subsequent deletion of the C-terminal domain. This mutation changed the localization of FMRP from cytoplasmic to nuclear. Further analyses in human cells, and using *Drosophila* as an *in vivo* model, demonstrated a novel and evolutionarily conserved role for FMRP C-terminus in nuclear export. This change in localization caused the gain of a novel function in neurons *in vivo*.

#### Impact
Our study suggests that several FXS patients may remain undiagnosed because clinics only screen for the epigenetic silencing of *FMR1* when FXS is suspected. Sequencing the coding region would be important to determine whether a patient has FXS, and may therefore benefit from future treatments. Furthermore, we identify the C-terminal domain of the FMRP as a nuclear export domain, perhaps explaining why naturally occurring nuclear isoforms of FMRP are generated by alternative splicing in the C-terminus.

primary antibodies using the standard protocol described in Soldano *et al* (2013). Primary antibodies anti-GFP (A-11222, Life Technologies, dil. 1:500), anti-βGal (A-11132, Life Technologies, dil. 1:1,000), anti-Dfmrp (20E4, specifically generated for our lab by EMBL-MACF Hybridoma, dil. 1:50) and Alexa Fluor secondary antibodies (Life Technologies) were used. Images of the stained fly brains were acquired using confocal microscopy, Nikon AIR confocal unit mounted on a TI2000 inverted microscope (Nikon Corporations).

## Quantification of axonal branching of LNv neurons

Maximum projections of the confocal stacks were analyzed using ImageJ software. During analysis, the experimenter was blinded to the genotypic conditions. Brains with gross mechanical damage from dissections/staining procedure were excluded from the analysis. A minimum sample size of 20 was analyzed for each genotype, deemed sufficient to detect changes in Dfmrp activity based on previous studies (Reeve *et al*, 2005, 2008). The branching areas of small LNv axonal termini were calculated based on parameters defined by Reeve *et al* (2005). The branching area was manually outlined, starting from the first point of defasciculation of the axonal bundle. Branching area measurements were normalized for variability in brain size across samples by dividing branching area values by total LNv commissure length for each brain. The Shapiro test for normality indicated that all were normally distributed ($P > 0.05$) except for the UAS-dFMR1$^{wt}$ sample, where the data, apart from one outlier, were also normally distributed. ANOVA indicated that genotype was a significant variable explaining variation in normalized

axon branching values ($F = 42.32$, $P < 0.0001$). Two-tailed *t*-tests using Welch's correction were then used to compare controls with mutant phenotypes. Similar results were obtained when a nonparametric Wilcoxon test was used (not shown). Normalized branching area values for each genotype were then analyzed based on a two-tailed *t*-test with Welch's correction, using GraphPad Prism software. Error bars represent mean values with SD.

**Supplementary information** for this article is available online:
http://embomolmed.embopress.org

## Acknowledgements

We thank Dr. Aaron New, Dr. Alessia Soldano, Simon Weinberger, other members of the Hassan lab and Canmert Koral for helpful discussions and comments on the manuscript. We would like to thank M. Baghdadi and Layka Abbasi for technical assistance. This work was supported by VIB (to BAH), the Belgian Science Policy Interuniversity Attraction Pole (BELSPO IUAP) networks (P7/20-WiBrain to BAH and P7/43-BeMGI to HVE, KD, and GF), Fonds Wetenschappelijke Onderzoeks (FWO) Grants G.0543.08, G.0680.10, G.0681.10, G.0682.10, and G.0503.12 (to BAH), by grants from the Geconcerteerde Onderzoeks Acties (GOA) of the University of Leuven (GOA/12/015 to HVE, KD, and GF), by the Netherlands Organization for Health Research and Development (RW; ZonMw; 912-07-022), and FRAXA Research Organization (to RW and FMSdV). HVE and KD are clinical investigators of the FWO. The Nikon AIR confocal used in the study was acquired through the Hercules Type 1 AKUL.09.037 grant.

## Author contributions

ZO, CEFdE, AC, JY, JV, GF, and FMSdV performed the experiments. HVE and KD provided clinical data and EBV cell lines. GF, FMSdV, RW, and BAH supervised the work. ZO and BAH wrote the manuscript with comments and edits from CEFdE, HVE, GF, FMSdV, and RW.

## Conflict of interest

The authors declare that they have no conflict of interest.

## For more information

Online Mendelian Inheritance in Man (OMIM), http://www.omim.org
Ensembl Genome Browser, http://www.ensembl.org
ScanProsite Tool provided by Expasy, http://prosite.expasy.org/scanprosite/
Leiden Open Variation Database (LOVD), http://www.lovd.nl/3.0/home

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
