## [Review Process File · EMBO Molecular Medicine]

A novel fragile X syndrome mutation reveals a conserved role for the carboxy-terminus in FMRP localization and function

Zeynep Okray, Celine E.F. de Esch, Hilde Van Esch, Koen Devriendt, Annelies Claeys, Jiekun Yan, Jelle Verbeeck, Guy Froyen, Rob Willemsen, Femke M.S. de Vrij, and Bassem A. Hassan

Corresponding author: Bassem Hassan, VIB Center for the Biology of Disease

Review timeline:

Submission date:	26 August 2014
Editorial Decision:	19 September 2014
Revision received:	19 December 2014
Accepted:	09 January 2015

Transaction Report:

Editor: Céline Carret

1st Editorial Decision

19 September 2014

Thank you for the submission of your manuscript to EMBO Molecular Medicine. We have now heard back from the three referees whom we asked to evaluate your manuscript.

You will see from the set of comments below that the three referees are overall supportive of publication, although they do require additional controls to increase conclusiveness, and suggest providing better justifications for the model used and maybe rephrase here and there.

Given these evaluations, I would be happy to receive a revised version of your manuscript, with the understanding that the referees' concerns must be fully addressed. Please note that it is EMBO Molecular Medicine policy to allow only a single round of revision and that, as acceptance or rejection of the manuscript may depend on another round of review, your responses should be as complete as possible.

I look forward to seeing a revised form of your manuscript as soon as possible.

***** Reviewer's comments *****

Referee #1 (Remarks):

I liked this paper very much. While the vast majority of patients with Fragile X Syndrome exhibit CGG repeat expansion, there are individuals with more subtle mutations. These have not been examined in as much detail and some may shed important light on molecular features and mechanisms of the FMRP protein. The current manuscript focuses on one such mutation, which causes a frameshift in the C-terminus of the protein. While this mutation appears to result in much lower levels of the FMRP protein in the patient (perhaps the real cause of the phenotype in this case), the mutant protein shows an unusual nucleolar localization. The authors characterize this in elegant detail and, using a variety of approaches including *Drosophila* genetics, demonstrate that the FMRP C-terminus appears to harbor a nuclear export signal which is functionally conserved. In my opinion, the experiments are solid, well presented and the manuscript is clearly written.

Referee #2 (Comments on Novelty/Model System):

The manuscript of Hassan et al. uses different approaches, from cell lines to a genetic model to tackle an important issue. This constitutes a strong and original point of the presented work.

Referee #2 (Remarks):

The manuscript from Hassan and collaborators deals with a very important topic: the physiopathological mechanisms underlying the Fragile X syndrome. Most patients display CGG repeat expansions at the FMR1 locus, which results in the loss of the FMR1 protein, however, there are patients that display the signs of the Fragile X syndrome but do not carry such defects. The authors present the molecular characterization of a specific FMR1 frame shift mutation that deletes the C terminal part and adds a nuclear localization signal. The study elegantly combines a variety of approaches and models starting from the mutation present in the patient and ending with flies. This allows the authors to draw new conclusions on the features and role of the FMR1 protein, including the presence of a nuclear export signal in the C terminal region and 3' to the RGG box.

I believe this is a nice piece of work that deserves publications provided the authors revise the manuscript along the following lines.

Can the author explicit what is the control used in Fig 1C?

It would be useful to know the amount of expressed wild type and mutant protein in the cell culture and in the fly experiments, as this may affect the interpretation. As an example, to demonstrate that the (DeltaCt+NLS) fly protein does not induce any significant phenotype in Fig 5D, the authors must eliminate the possibility that the amounts of produced proteins are not comparable to those of the other transgenic lines.

Given that the C region is not very well conserved, the use of the fly model for this particular mutation should be better justified. Information concerning sequence conservation in the C terminal region should be provided, including the information on the precise insertion engineered in flies. This will strengthen the argument that flies can be used to study human proteins.

The nucleolar localization of the mutant protein is clear in HEK293 and Sy5y cells but not in neurons. Since neurons are considered the main site of action of FMR1, the authors should be more cautious about their conclusions and play down the interpretation. It is also possible that the evoked mechanism (nucleolar accumulation) is correct and that the neuronal population used in the study does not show the strongest effect. In this case, the authors should try another neuronal population.

Typos:

NLS instead of NoS in Fig 4A.

In flies, genes are indicated by lower case and in italic, proteins by upper case for the first letter and not in italic (see for example page 2 *dfmrp*).

Page 3 first line second paragraph, I guess it should read FMRP is an RNA-binding protein THAT regulates...

Page 12 'high functioning males' should be modified.

Referee #3 (Remarks):

In MS "A novel fragile X syndrome mutation reveals a conserved role for the carboxy-terminus in FMRP localization and function" Okray and colleagues isolate a novel FMRP mutation from an FRX patient. This mutation causes a frameshift, and in cell culture causes nuclear accumulation of the protein. In flies, overexpression of the protein causes novel phenotypes, leading the authors to conclude that this allele behaves as a neomorph.

The paper is clearly written and describes an interesting novel mutation associated with FRX. With some additional evidence supporting (or rejecting) the claim for neomorphism, the paper is well suited for EMBO Mol Med.

Major issue:

The authors claim that the novel mutant, FMRP[G-ins], behaves in a neomorphic way based on its unusual nuclear localization, and phenotypes when expressed in flies. At the same time they mention that FMRP has actually been observed in the nucleolus. If that is true, the cell culture data wouldn't really argue for a neomorphic function, but just an increase of normally found nucleolar function. It might help the authors to settle this with an experiment along the lines of Figure 4: Express wild-type FMRP, and add LMB. Will the normal form of FMRP be found in the nucleus or not?

Similarly, the branching phenotype in flies could be an extreme hypermorphic phenotype (i.e. much more normal FMR activity). If FMR[dCT+NLS] was really a neomorph, the additional expression of wild-type FMR should have no effect on the occurrence and frequency of axon looping. (The classical definition of a neomorphic mutation being that its phenotype is impervious to the amount of wild type copies in the background). Together, these data would strengthen the conclusions about the nature of this allele. As would quantification of the above mentioned cell culture, and neuronal phenotypes.

Minor:

The abstract jumps from epigenetics to CGG repeats without explaining the correlation to the non-initiated.

The RGG box label is almost illegible in the figures. And Figure 1A would benefit from a wild type schematic.

Also, the combination of a cylindrical protein schematic, with straight deletion lines makes these schematics somewhat unintuitive.

1st Revision - authors' response

19 December 2014

We thank the reviewers for their positive comments on our work. We have addressed their concerns with new data. Below we detail the new data and the textual changes we made to the manuscript to address the referees' concerns.

Referee #1:

I liked this paper very much. While the vast majority of patients with Fragile X Syndrome exhibit CGG repeat expansion, there are individuals with more subtle mutations. These have not been

examined in as much detail and some may shed important light on molecular features and mechanisms of the FMRP protein. The current manuscript focuses on one such mutation, which causes a frameshift in the C-terminus of the protein. While this mutation appears to result in much lower levels of the FMRP protein in the patient (perhaps the real cause of the phenotype in this case), the mutant protein shows an unusual nucleolar localization. The authors characterize this in elegant detail and, using a variety of approaches including Drosophila genetics, demonstrate that the FMRP C-terminus appears to harbor a nuclear export signal which is functionally conserved. In my opinion, the experiments are solid, well presented and the manuscript is clearly written.

Response

We are grateful to the referee for these very positive comments.

Referee #2:

The manuscript of Hassan et al. uses different approaches, from cell lines to a genetic model to tackle an important issue. This constitutes a strong and original point of the presented work. The manuscript from Hassan and collaborators deals with a very important topic: the physiopathological mechanisms underlying the Fragile X syndrome.

Most patients display CGG repeat expansions at the FMR1 locus, which results in the loss of the FMR1 protein, however, there are patients that display the signs of the Fragile X syndrome but do not carry such defects. The authors present the molecular characterization of a specific FMR1 frame shift mutation that deletes the C terminal part and adds a nuclear localization signal. The study elegantly combines a variety of approaches and models starting from the mutation present in the patient and ending with flies. This allows the authors to draw new conclusions on the features and role of the FMR1 protein, including the presence of a nuclear export signal in the C terminal region and 3' to the RGG box.

Response

We are grateful to the referee for these very positive comments.

I believe this is a nice piece of work that deserves publications provided the authors revise the manuscript along the following lines.

1- Can the author explicit what is the control used in Fig 1C?

Response

The control in figure 1C is one of several unaffected individuals from whom we had cell lines available. We have amended the sentence on page 7 as follows:

Western blot analysis indicated that FMRP protein levels in patient-derived cells were significantly decreased compared to control cells from a healthy individual with no CGG repeat expansion in the FMR1 locus (Fig 1D).

2- It would be useful to know the amount of expressed wild type and mutant protein in the cell culture and in the fly experiments, as this may affect the interpretation. As an example, to demonstrate that the (DeltaCt+NLS) fly protein does not induce any significant phenotype in Fig 5D, the authors must eliminate the possibility that the amounts of produced proteins are not comparable to those of the other transgenic lines.

Response

This is an excellent point. In cell culture experiments, the same expression vector and transfection parameters were used for all the FMR1 variants. In the fly experiments, we inserted all the transgenes in the same genomic locus, to reduce positional effects on gene expression. While these would normalize transcriptional effects, it does not guarantee equal levels of protein expression across the different alleles, especially given our observation that FMRP levels are post-transcriptionally reduced in patient-derived cells. To test the protein levels of the different human and fly FMR1 variants used, we performed western blot analyses. We find that the protein levels are indeed quite different, with the patient mutation, and its fly mimic exhibiting *lower* expression levels than the wild type protein and the C-terminal truncation alone, but comparable levels to the

controls where the NLS is mutated or the C-terminus is restored. In the fly experiment, all controls - including the lower expressed Dfmrp^{wt+NLS} - cause axonal collapse, suggesting that the failure of Dfmrp^{ΔCt+NLS} expression to cause collapse is not merely a consequence of lower protein levels. Furthermore, the change in localization of the patient protein in human cells and in flies, and the de novo axonal phenotypes, do not correlate with expression levels and are certainly not due to increased levels of protein expression. In other words, despite being expressed at lower levels than the wild type protein, the patient mutation causes changes in localization and the novel axonal phenotypes in vitro and in vivo, respectively.

We have added these protein level analyses as Supplementary Fig 2B and Supplementary Fig 4 and added the following sentences:

(Page 9) Importantly, the change in subcellular localization of the patient protein compared to the various controls does not correlate with levels of protein expression (Supplementary Fig 2B). Specifically, the patient mutation exhibits lower expression levels than the wild type protein and the C-terminal truncation alone, but comparable levels to the controls where the NLS is mutated or the C-terminus is restored. Together, these data suggest that the nucleolar localization is due to the mutation and not to an increase in expression level.

(Page 12) The de novo axonal phenotypes appear despite the fact that Dfmrp^{ΔCt+NLS} is expressed at lower levels than the wild type or Dfmrp^{ΔCt} controls, and at comparable levels to the Dfmrp^{wt+NLS} control (Supplementary Fig 4), consistent with the data for the human counterparts (Supplementary Fig 2B). Together, these data suggest that the defects caused by the Dfmrp^{ΔCt+NLS} are an intrinsic property of the compound mutations, rather than expression levels of the mutant protein.

3- Given that the C region is not very well conserved, the use of the fly model for this particular mutation should be better justified. Information concerning sequence conservation in the C terminal region should be provided, including the information on the precise insertion engineered in flies. This will strengthen the argument that flies can be used to study human proteins.

Response

We have added sequence comparison information as the new supplementary figure 3 and referred to that on page 10. In this figure, we have also indicated the precise location at which the truncation was engineered in Dfmrp^{ΔCt} and Dfmrp^{ΔCt+NLS} variants.

In terms of justifying the use of the fly model, the original manuscript contained a paragraph on page 10, which explains this and references several studies on fly *dfmr1*. We have now made this more explicit in the introduction on page 5 by adding and referencing the following sentence:

A genetically versatile model for *FMRI* loss of function has been established in the fruit fly *Drosophila melanogaster*. The fly has a single homologue of *FMRI*, and its loss of function causes many phenotypes similar to those associated with FXS patients

Finally, it should be noted that it is precisely because the C-terminus is not highly conserved that we questioned whether the nuclear export function is conserved or human-specific. In that sense, the lack of C-terminus conservation in the fly protein is what makes it a valuable model to ask about functional conservation in protein localization. The original manuscript states this on page 10.

*4- The nucleolar localization of the mutant protein is clear in HEK293 and Sy5y cells but not in neurons. Since neurons are considered the main site of action of *FMRI*, the authors should be more cautious about their conclusions and play down the interpretation. It is also possible that the evoked mechanism (nucleolar accumulation) is correct and that the neuronal population used in the study does not show the strongest effect. In this case, the authors should try another neuronal population.*

Response

The nucleolar localization in primary mouse neurons is clearly visible, but we agree that it is significantly weaker than in cell lines. Nonetheless, the reviewer is right as we cannot be sure that the change in localization we observe in human and mouse cells is also true in the patient brain.

In order to be cautious in all our interpretations, we have removed the term “neomorphic” from the abstract and several other – but not all – places in the manuscript, and also added the following cautionary note on page 13 in the discussion section:

Future experiments at endogenous expression levels *in vivo* will help ascertain whether these interpretations are true.

5- Typos:

NLS instead of NoS in Fig 4A.

*In flies, genes are indicated by lower case and in italic, proteins by upper case for the first letter and not in italic (see for example page 2 *dfmrp*).*

Page 3 first line second paragraph, I guess it should read FMRP is an RNA-binding protein THAT regulates...

Page 12 'high functioning males' should be modified.

Response

All typos have been corrected.

Referee #3:

In MS "A novel fragile X syndrome mutation reveals a conserved role for the carboxy-terminus in FMRP localization and function" Okray and colleagues isolate a novel FMRP mutation from an FRX patient. This mutation causes a frameshift, and in cell culture causes nuclear accumulation of the protein. In flies, overexpression of the protein causes novel phenotypes, leading the authors to conclude that this allele behaves as a neomorph.

The paper is clearly written and describes an interesting novel mutation associated with FRX. With some additional evidence supporting (or rejecting) the claim for neomorphism, the paper is well suited for EMBO Mol Med.

Response

We are grateful to the referee for these very positive comments.

Major issue:

1- The authors claim that the novel mutant, FMRP [G-ins], behaves in a neomorphic way based on its unusual nuclear localization, and phenotypes when expressed in flies. At the same time they mention that FMRP has actually been observed in the nucleolus. If that is true, the cell culture data wouldn't really argue for a neomorphic function, but just an increase of normally found nucleolar function. It might help the authors to settle this with an experiment along the lines of Figure 4: Express wild-type FMRP, and add LMB. Will the normal form of FMRP be found in the nucleus or not?

Response

This is an excellent point. We had failed in the original manuscript to cite the study by Dury and colleagues (Plos Genetics, 2013) which shows that full length wild type FMRP does not localize to the nucleus/nucleolus even upon LMB treatment. We have added the following sentence on page 9:

It has been shown that full-length wild type FMRP does not show nuclear retention upon LMB treatment (Dury et al., 2013).

We ascertained for ourselves that this is indeed the case and include a figure here for the referee's review. We did not add these data to the manuscript, as they are already published.

We conclude that the nuclear localization of the patient allele is indeed a novel characteristic not shared by the full-length wild type protein. As we stated in our original discussion however, one specific shorter isoform of endogenous FMRP does seem to show nuclear localization upon LMB treatment. This is why we suggested in the original manuscript that the patient mutation may represent a mimic of a “reduction in diversity” of FMRP isoforms. However, as we now state in the revised discussion (page 14), *in vivo* endogenous expression levels should be examined in the future to ascertain this.

2- Similarly, the branching phenotype in flies could be an extreme hypermorphic phenotype (i.e. much more normal FMR activity). If FMR1[dCT+NLS] was really a neomorph, the additional expression of wild-type FMR should have no effect on the occurrence and frequency of axon looping. (The classical definition of a neomorphic mutation being that its phenotype is impervious to the amount of wild type copies in the background). Together, these data would strengthen the conclusions about the nature of this allele. As would quantification of the above mentioned cell culture, and neuronal phenotypes.

Response

This is a very valid concern, also raised by referee2. In the fly experiments, we inserted all the transgenes in the same genomic locus, to reduce positional effects on gene expression. While these would normalize transcriptional effects, it does not guarantee equal levels of protein expression across the different alleles, especially given our observation that levels of the patient protein and some of the other relevant variants are different compared to wt FMRP. To test the protein levels of the different human and fly *FMR1* variants used, we performed western blot analyses. We find that the protein levels are indeed quite different, with $Dfmrp^{\Delta CT+NLS}$ are exhibiting *lower* expression levels than the wild type protein and the C-terminal truncation alone, but comparable levels to the control where the C-terminus is restored ($Dfmrp^{wt+NLS}$). Thus, the nuclear localization of the patient-like protein in flies, and the de novo axonal phenotypes, do not correlate with expression levels and are certainly not due to increased levels of protein expression. In other words, despite being expressed at lower levels than the wild type protein, the patient mutation causes changes in localization and results in novel axonal phenotypes.

Similarly, in cell culture experiments, the change in subcellular localization of the patient protein compared to the various controls does not correlate with levels of protein expression (Supplementary Fig 2B). Specifically, the patient mutation exhibits lower expression levels than the wild type protein and the C-terminal truncation alone, but comparable levels to the controls where the NLS is mutated or the C-terminus is restored. These data suggest that the nucleolar localization is due to the mutation and not to an increase in expression level.

We have added these protein level analyses as Supplementary Fig 2B and Supplementary Fig 4 and added the following sentences:

(Page 9) Importantly, the change in subcellular localization of the patient protein compared to the various controls does not correlate with levels of protein expression (Supplementary Fig 2B). Specifically, the patient mutation exhibits lower expression levels than the wild type protein and the C-terminal truncation alone, but comparable levels to the controls where the NLS is mutated or the C-terminus is restored. Together, these data suggest that the nucleolar localization is due to the mutation and not to an increase in expression level.

(Page 12) The de novo axonal phenotypes appear despite the fact that Dfmrp^{ΔCt+NLS} is expressed at lower levels than the wild type or Dfmrp^{ΔCt} controls, and at comparable levels to the Dfmrp^{wt+NLS} control (Supplementary Fig 4), consistent with the data for the human counterparts (Supplementary Fig 2B). Together, these data suggest that the defects caused by the Dfmrp^{ΔCt+NLS} are an intrinsic property of the compound mutations, rather than expression levels of the mutant protein.

Furthermore, we compared the effects of expressing 1 wild type *dfmr1* to 2 wild type *dfmr1* transgenes (*UAS-dfmr1^{wt}* vs. *UAS-dfmr1^{wt} x2*) on the axonal phenotype of LNV neurons. In parallel, we also attempted to compare the effects of expressing *dfmr1^{ΔCt+NLS}* alone to expressing *dfmr1^{ΔCt+NLS}* together with *dfmr1^{wt}* (*UAS-dfmr1^{ΔCt+NLS}* vs. *UAS-dfmr1^{wt}+UAS-dfmr1^{ΔCt+NLS}*). Increasing the amount of wild type dFMRP expression (*UAS-dfmr1^{wt}* vs. *UAS-dfmr1^{wt} x2*) did not cause nuclear localization nor axonal looping or bifurcation. These data are now presented as supplementary figure 5 and described on page 12. It was not possible to reliably assess axonal phenotypes in LNV neurons of flies co-expressing *dfmr1^{ΔCt+NLS}* with *dfmr1^{wt}* as the co-expression caused cell lethality and axonal degeneration, which we never see with the expression of either allele alone. It should be emphasized here that Dfmrp^{ΔCt+NLS} is expressed at significantly lower levels than the wild type counterpart. We therefore believe that a neomorphic interpretation is more consistent with the data. Nonetheless, because these observations rely on the expression in transgenes rather than study of endogenous alleles, we have replaced the term “neomorphic” with the term “novel” in the abstract and at several – but not all – positions throughout the manuscript.

The quantification of the penetrance of novel axonal phenotypes in LNV neurons of flies expressing *dfmr1^{ΔCt+NLS}* is now included in Fig 6. The nucleolar localization of the patient FMRP in our cell culture experiments is highly penetrant, occurring in almost all cells studied.

Minor:

1- The abstract jumps from epigenetics to CGG repeats without explaining the correlation to the non-initiated.

Response

This has been clarified in the abstract.

2- The RGG box label is almost illegible in the figures. And Figure 1A would benefit from a wild type schematic. Also, the combination of a cylindrical protein schematic, with straight deletion lines makes these schematics somewhat unintuitive.

Response

This has been rectified in the new figures.